# Open versus Minimally Invasive Partial Nephrectomy: Trends and Outcomes from a Wide National Population-Based Database

**DOI:** 10.3390/jcm13185454

**Published:** 2024-09-14

**Authors:** Antonio Franco, Riccardo Lombardo, Francesco Ditonno, Eugenio Bologna, Leslie Claire Licari, Omar Nabulsi, Darren Ioos, Giacomo Gallo, Giorgia Tema, Antonio Cicione, Antonio Nacchia, Andrea Tubaro, Cosimo De Nunzio, Edward E. Cherullo, Riccardo Autorino

**Affiliations:** 1Department of Urology, Rush University Medical Center, Chicago, IL 60612, USA; anto.franco@hotmail.it (A.F.); ricautor@gmail.com (R.A.); 2Department of Urology, Sant’Andrea Hospital, La Sapienza University, 00185 Rome, Italy; giacomo.gallo@uniroma1.it (G.G.); acicione@libero.it (A.C.);; 3Department of Urology, University of Verona, 37129 Verona, Italy; 4Urology Unit, Department of Maternal-Child and Urological Sciences, “Sapienza” University of Rome, Policlinico Umberto I Hospital, 00161 Rome, Italy

**Keywords:** RAPN, minimally invasive surgery, enucleation, trends, SDOH

## Abstract

**Objectives:** To investigate temporal trends and overall complication rates among open partial nephrectomy (OPN) and minimally invasive partial nephrectomy (MIPN), including the impact of social determinants of health (SDOH) on postoperative outcomes. **Methods:** Patients who underwent OPN or MIPN between 2011 and 2021 were retrospectively analyzed by using PearlDiver-Mariner, an all-payer insurance claims database. The International Classification of Diseases diagnosis and procedure codes were used to identify the type of surgical operation, patient’s characteristics (age, sex, region, insurance plan), postoperative complications and SDOH, categorized in education, healthcare, environmental, social, and economic domains. Outcomes were compared using multivariable regression models. **Results:** Overall, 65,325 patients underwent OPN (n = 23,377) or MIPN (n = 41,948). OPN adoption declined over the study period, whereas that of MIPN increased from 24% to 34% (*p* = 0.001). The 60-day postoperative complication rate was 15% for the open and 9% for the minimally invasive approach. Approximately 16% and 11% of patients reported at least one SDOH at baseline for OPN and MIPN, respectively. SDOH were associated with higher odds of postoperative complications (OPN = OR: 1.11, 95% CI: 1.01–1.25; MIPN = OR: 1.31, 95% CI: 1.18–1.46). The open approach showed a significantly higher risk of postoperative complications (OR: 1.62, 95% CI: 1.54–1.70) compared to the minimally invasive one. **Conclusions:** Our findings confirm that MIPN is gradually replacing OPN, which carries a higher risk of complications. SDOH are significant predictors of postoperative complications following PN, regardless of the approach.

## 1. Introduction

Current European and American guidelines recommend partial nephrectomy (PN) as the standard surgical treatment for clinical T1a renal tumors whenever technically feasible [1,2]. Over time, several minimally invasive techniques for the treatment of kidney tumors have been developed, including laparoscopic partial nephrectomy (LPN), robot-assisted partial nephrectomy (RAPN) and ablative techniques [3,4,5,6,7]. With the patient in the flank position, these techniques utilize sub-centimetric incisions for trocar placement to access the peritoneal cavity and renal hilum instead of the classic anterior subcostal incision below the 11th rib used in open surgery. The subsequent steps resemble those of an open partial nephrectomy (OPN) [8]. These approaches proved to be effective, safe and non-inferior to the traditional open approach, leading to their gradual increased adoption, even for more advanced and complex masses [9]. More specifically, evidence suggests fewer overall complications, fewer major complications, fewer transfusions and a much shorter hospital stay for the minimally invasive surgery (MIS) [9,10,11]. However, despite the trend towards minimally invasive surgery, large-scale data supporting its use and benefits are still lacking.

Furthermore, impaired access to minimally invasive surgery has been reported among ethnic minorities and socioeconomically disadvantaged groups [12,13], leading to worse oncological outcomes in these populations [14]. More specifically, social determinants of health (SDOH) are recognized as a significant source of disparity affecting perioperative outcomes in various major surgeries [15,16] including renal cell carcinoma (RCC) [7]. Based on the United States’ Healthy People 2030 initiative, they can be categorized into 5 domains: education, healthcare, environmental, social, and economic [17,18].

High-quality evidence regarding surgical trends and PN approaches adoption across different socioeconomic groups and regions of the United States is limited. Therefore, the aim of our study is to provide a comprehensive nationwide population-based analysis of trends of PN for RCC.

## 2. Materials and Methods

### 2.1. Data Set

The PearlDiver Mariner (PearlDiver Technologies, Colorado Springs, CO, USA) is a commercially available, Health Insurance Portability and Accountability Act (HIPAA)-compliant, de-identified U.S. database of insurance billing records. This database encompasses records of all healthcare encounters billed in both inpatient and outpatient settings, enabling longitudinal tracking of patients over time. It includes data on medical and surgical claims for over 150 million unique patients collected from 2011 to 2021. A retrospective observational cohort analysis was conducted using this database [19]. The primary objective was to evaluate trends of PN for RCC in a large population-based database. As part of a secondary analysis, overall complication rates and various sociodemographic variables for both MIPN and OPN were assessed, including the impact of SDOH on perioperative outcomes. Given the use of de-identified data, the study was granted exempt status from institutional review board oversight.

### 2.2. Study Population

Patients with RCC diagnoses were identified by using appropriate International Classification of Diseases (ICD)-9 and ICD-10 diagnosis codes. Then, partial nephrectomy ICD-9/10 procedure codes and current procedural terminology (CPT) codes were matched with RCC diagnosis codes in order to identify the precise cohort (Appendix A). All patients with a first diagnosis of RCC who underwent PN were included in the analysis. Furthermore, the cohort was restricted to subjects with active insurance claims data or those who made cash payments for claims in the year preceding or the year following the initial diagnosis. Differentiation between laparoscopic and robot-assisted surgery was not feasible due to the absence of specific procedural codes for each approach. The overall population was stratified into OPN and MIPN according to treatment modality. For each patient, age, sex, region, Charlson comorbidity index (CCI), insurance type, length of stay and year of surgery variables were extracted. SDOH were evaluated using Z-codes from the ICD-9/10, as detailed in Appendix A. Overall complications were defined within 60 days after surgery and required at least one of the conditions listed in Appendix A.

### 2.3. Statistical Analysis

The distributions of patient characteristics were described using means (±SD) and patient counts with percentages. A two-sided *t*-test and chi-square test were employed to compare continuous and categorical variables between different treatment modalities. A Cochran-Armitage test was used to perform temporal analysis of surgical trends between the OPN vs. MIPN approaches. A multivariable logistic regression analysis was carried out to identify predictors of perioperative complications, adjusting for age, sex, CCI, year of surgery, region, insurance type, SDOH and type of approach.

Analyses were conducted using R statistical language integrated with the Bellwether software (PearlDiver Technologies Inc., Colorado Springs, CO, USA), connected to the Mariner database [19], with statistical significance set at *p* < 0.05.

## 3. Results

### 3.1. Demographics

Overall, 65,325 patients underwent OPN (n = 23,377, 36%) or MIPN (n = 41,948, 64%) during the study period. Mean (±SD) age was 60.6 yrs (±14.3) and 61.5 yrs (±11.8) for OPN and MIPN approaches, respectively. The mean (±SD) CCI was 4.00 (±3.29) and 4.48 (±2.85) (*p* < 0.001) and 79% vs. 75% of patients had a CCI ≥ 3 (*p* < 0.001) for OPN and MIPN approaches, respectively. PN was mostly performed in the Southern region (24,083, 37%), while the lowest number of procedures was observed in the West (8173, 12%). The insurance regimen was similar between the two approaches, although a significant difference was observed (*p* < 0.001), with most of the overall cohort privately insured (44,991, 69%). Regarding SDOH, 8354 (13%) patients in the whole cohort presented at least one to maximum five SDOH domains (Table 1).

### 3.2. Temporal Trends

The adoption of the open approach slightly declined across the study period (from 32% in 2011–2014 to 26% in 2019–2021), whereas MIPN utilization increased from 24% to 34%. The proportion of MIPN as a fraction of the whole cohort increased significantly over time, reflecting a statistically significant rising trend (slope of regression line, reg. = 0.07, *p* = 0.001). Conversely, proportions for OPN cases (reg. = −0.07, *p* = 0.001) decreased (Figure 1).

### 3.3. Complications and SDOH

Open surgery for RCC was exposed to a higher rate of perioperative complications, whereas the MIPN cohort more frequently had an uncomplicated postoperative course, with an overall complications rate of 15% vs. 9% (*p* < 0.001), respectively. In terms of specific types of complications, higher rates of blood transfusion, acute kidney injury (AKI) and leakage were observed in the OPN group (*p* < 0.001), whereas no significant differences were noted between the two approaches for ileus, vascular injuries, sepsis or pulmonary embolism complications (Table 2).

Moreover, OPN had longer LoS (4.78 ± 2.34 days vs. 3.12 ± 1.82 days *p* < 0.001) and more frequently experienced hospital readmission within 30 days from the surgical operation compared to MIPN (8% vs. 5%, *p* < 0.001).

Regarding SDOH, 3740 (16%) patients who underwent OPN were identified with z-codes indicating the presence of SDOH, whereas only 4614 (11%) were observed in the MIPN group (*p* < 0.001).

### 3.4. Multivariable Analysis

At multivariable analysis, age (OR 1.01, 95% CI 1.01–1.02) and CCI (OR 1.10, 95% CI 1.09–1.11) were significant predictors of perioperative complications across the model. Regarding type of technique, patients undergoing OPN had significantly higher probabilities of experiencing perioperative complications (OR 1.62, 95% CI 1.54–1.70) compared to MIPN. Finally, the presence of SDOH was associated with higher odds of postoperative complications for both OPN (OR 1.11, 95% CI 1.01–1.25) and MIPN approaches (OR 1.31, 95% CI 1.18–1.46) (Table 3).

## 4. Discussion

Our national population-based study investigated sociodemographic characteristics, temporal trends, and perioperative complications related to PN, including analysis of potential sources of disparity that could affect surgical outcomes.

Despite long-standing evidence indicating equivalent oncological outcomes and improved postoperative benefits for laparoscopic and robotic renal surgery [9,10,11,20], current guidelines still do not favor MIS over open surgery for kidney cancer management [1,2]. Undoubtedly, a major shift towards minimally invasive treatment for localized RCC has been witnessed. A national population-based study by Banegas et al. highlighted a significant shift toward the increased use of nephron-sparing and minimally invasive surgical techniques to treat RCC patients in the U.S., showing respectively that among patients with stage I tumors ≤4 cm and >4–7 cm, the use of PN significantly increased from 43% in 2004 to 55% in 2009 (*p* ≤ 0.05) and LPN increased from 8% to 15% [21]. Similarly, Weight et al. reported that approximately 5 to 7 years after the introduction of LPN and laparoscopic radical nephrectomy (RN), these MIS techniques equaled and then surpassed the number of open PN and RN in a 10-year single center experience [22]. The widespread adoption of MIPN has certainly been facilitated by the introduction of robotic platforms in the field of renal oncological surgery, as evidenced by various studies [23,24,25]. For instance, Patel et al. reported a rising trend for RAPN from 2008 to 2011, attaining a 14% rate at university and a 10% rate at non-university hospitals (*p* = 0.03), suggesting that robotic technology may enable surgeons across practice settings to more frequently perform nephron-sparing surgery [23]. In our study, the utilization of MIPN increased over 10 years’ time, reflecting a statistically significant rising trend (slope of regression line, reg. = 0.07, *p* = 0.001) compared to OPN. A slight decrease for both procedures was observed in 2019–2021. A possible explanation may be related to the outbreak of the COVID-19 pandemic, which profoundly impacted access to healthcare, resulting in more frequent non-surgical management for renal masses.

Regarding perioperative complications, our analysis showed a higher rate of perioperative complications when using the open technique, whereas the MIPN cohort more frequently had an uncomplicated postoperative course (15% vs. 9%, *p* < 0.001), thus leading to a shorter LoS (3.12 ± 1.82 days vs. 4.78 ± 2.34 days, *p* < 0.001) and a lower readmission rate within 30 days (5% vs. 8%, *p* < 0.001).

These results reflect what the guidelines declare, namely, improved perioperative outcomes such as reduced blood loss, need of transfusion or shorter recovery for the MIPN approach [1,2]. However, most of the evidence relies on short follow-up outcomes and retrospective data. Guglielmetti et al. performed a prospective, randomized trial comparing the outcomes of open vs. LPN showing that the latter was associated with less bleeding and fewer transfusions, whereas OPN patients experienced more abdominal wall complications. Moreover, the laparoscopy group reported lower kidney function reduction at 3 and 12 months after surgery and a lower rate of down-staging on chronic kidney disease classification at 12 months, despite the fact that the trial was not powered for kidney function [26]. On the other hand, Marszalek et al. found that glomerular filtration rate (GFR) decline was greater in the LPN group in the immediate postoperative period, probably due to ischemia time or the negative impact of capnoperitoneum, but not after 3.6 years follow-up [27]. Although the PearlDiver database does not provide specific GFR and creatinine data to accurately quantify renal function variation, we did observe a higher incidence of AKI in the OPN group (4% vs. 2%, *p* < 0.001). However, the absence of long-term GFR, ischemia time, and on/off clamp technique data limits the generalizability of our results.

Studies investigating RAPN perioperative outcomes have rapidly grown during the last ten years, yet the impact of the robotic approach is still considered controversial by EAU guidelines [2].

The ROBOCOP II study is the first and only randomized control trial which succeeded in comparing open vs. RAPN, evaluating the feasibility of trial recruitment and the perioperative and oncological outcomes as a secondary endpoint. So far, an overall complication difference estimate rate of 40% was described, although it was mainly driven by minor complications favoring RAPN, with no significant differences in major complications between the two groups [11].

In line with our results, Zeuschner et al. retrospectively analyzed 500 RAPN vs. 313 OPN cases, recording an overall complication rate of 35% vs. 24% (*p* < 0.001), respectively; after propensity match scored analysis for tumor size and mass complexity, outcomes still favored the RAPN approach, although differences in estimated blood loss (EBL) were no longer significant [20].

It is noteworthy that our analysis could not extrapolate data regarding tumor size, nephrometry score and clinical stage due to intrinsic limitations of the database, and therefore our findings may not draw definitive conclusions. Nonetheless, at multivariable analysis, OPN had a significantly higher probability of perioperative complications (OR 1.62, 95% CI 1.54–1.70) than MIPN, corroborating the evidence suggesting a higher likelihood of experiencing a complicated perioperative course with this approach. Furthermore, a less complicated perioperative course may mean shorter LoS, as suggested in our analysis, and hence a lower healthcare expenditure. It is well-known that MIS usually retains higher costs in terms of surgery expenditure (instruments, platforms, etc.) [28]. However, according to Okhawere et al., 1-year healthcare expenditure after discharge for MIPN was not significantly different when compared to OPN, resulting in comparable total expenditures for the index period and 1-year post-discharge healthcare uses [29].

A growing body of studies focusing on how social factors influence health has led to a consistent increase of the literature on this topic [16,30,31]. Key domains of SDOH include economic stability, education, access to healthcare, neighborhood and environment, and social and community context [17,18]. These determinants are widely acknowledged as affecting people’s health both directly and indirectly, and they help to explain the relationship between socioeconomic conditions and individual health. Therefore, they can provide direction for policies, community initiatives, and interventions that may reduce the burden of disease [32]. In the context of kidney cancer, it has been shown that uninsured patients and Medicare or Medicaid recipients were less likely to receive MIS as compared to privately insured patients [33]. Indeed, this phenomenon also applies to racial and ethnic minorities, particularly in the setting of private health care [34]. As a matter of fact, worse oncological outcomes were reported among ethnic minorities and sociodemographic groups at lower socioeconomic status due to impaired access to MIS [14]. For instance, a recent review examining the relationship between social disparities and surgical outcomes discovered that patients with SDOH face increased risks of presenting a more advanced-stage disease, thus requiring emergency care, and of receiving treatment at low-volume institutions [32]. Moreover, Becker et al. investigated the presence of sociodemographic disparities in the treatment of small renal masses. Interestingly, socioeconomic status was higher for patients treated with PN (51.1 vs. 49.8%, *p* = 0.046), and patients with a higher socioeconomic status were significantly more likely to be treated with PN than with RN with respect to the percentage of persons below the poverty line and to family income. Furthermore, at multivariable analysis, black patients were more likely to be treated non-surgically than other ethnic groups, and men were more likely to be non-surgically managed compared to women. Overall, for every 10% increase in poverty, a 0.2% higher rate of non-surgical management was detected (*p* = 0.006) [35].

Our group has also previously investigated the impact of SDOH among minimally invasive PN, RN and renal ablation surgical outcomes. According to our findings, patients with SDOH had higher odds of experiencing perioperative complications irrespective of the adopted approach [7]. In the present study, we confirmed these results, as the presence of SDOH appeared to be a clinically relevant factor that affected outcomes of both OPN (OR 1.11, 95% CI 1.01–1.25) and MIPN (OR 1.31, 95% CI 1.18–1.46) approaches.

When evaluating social disparities in large national databases such as the PearlDiver Mariner, it is important to also consider insurance coverage trends. As a matter of fact, almost 70% of the PN cohort was privately insured, whereas only 5% of the population was covered by a Medicaid plan, thus reasonably reflecting the minority of SDOH patients across the overall study. Indeed, it is well-established that states incentivize Medicaid to address social needs by partnering with social services and community-based programs [36]. More specifically, these interventions and tailored programs include the following: increasing funding for healthcare facilities in underserved areas and mandating comprehensive insurance coverage that minimizes out-of-pocket costs for vulnerable populations; conducting programs that focus on education and early intervention in communities disproportionately affected by SDOH, improving awareness of surgical options and outcomes, and providing support systems that address barriers to accessing timely surgical care; and establishing patient navigator services in hospitals that assist individuals from disadvantaged backgrounds in navigating the healthcare system, ensuring they receive the necessary preoperative and postoperative care, and connecting them with social services to address broader SDOH-related challenges [18].

In summary, a solid body of evidence suggests an established shift towards minimally invasive surgery for PN, although long-term data and RCTs are still warranted to globally confirm MIPN’s safety and routine adoption.

The present study is not devoid of limitations. Firstly, despite careful code selection for data extraction, coding errors or misclassification could be possible. Additionally, the absence of distinct coding for laparoscopic and robot-assisted procedures hindered our ability to differentiate between the two approaches. However, it is noteworthy that the impact of this limitation on the validity of our results is mitigated by the fact that both CPT and ICD codes are nationally and internationally standardized.

Furthermore, it is pivotal to recognize that our study is based on a U.S. population, which limits the generalizability of our findings to this specific context. Additionally, most of the commercial claims in the database come from Humana and United Healthcare, which are more prevalent in Southeast regions. Consequently, despite the substantial sample size, there may be some level of selection bias that cannot be completely mitigated. Another intrinsic limitation of the PearlDiver Mariner database is the absence of data concerning patients’ race, tumor characteristics, size or nephrometry score, facility-related factors, lack of complications’ grading, oncological outcomes, information on long-term renal function data and operative outcomes such as ischemia time, operative time and estimated blood loss. Notwithstanding this limitation, we believe that our study provides a comprehensive overview of trends, outcomes and social disparities between different approaches for PN in the U.S. Furthermore, an extensive study on the impact of each SDOH domain on major urological surgery is ongoing, aiming to better clarify the role of these factors in the urological panorama and to overcome the intrinsic limitations of the PearlDiver database.

## 5. Conclusions

The present study highlights national trends related to the progressive increase in the utilization of MIPN. Our findings confirm that MIPN is gradually replacing OPN, which carries a higher risk of complication. SDOH are significant predictors of postoperative complications following PN regardless of the approach. Social services and community programs, as well as focused research on the topic, should try to mitigate their impact in the near future.

## Figures and Tables

**Figure 1 jcm-13-05454-f001:**
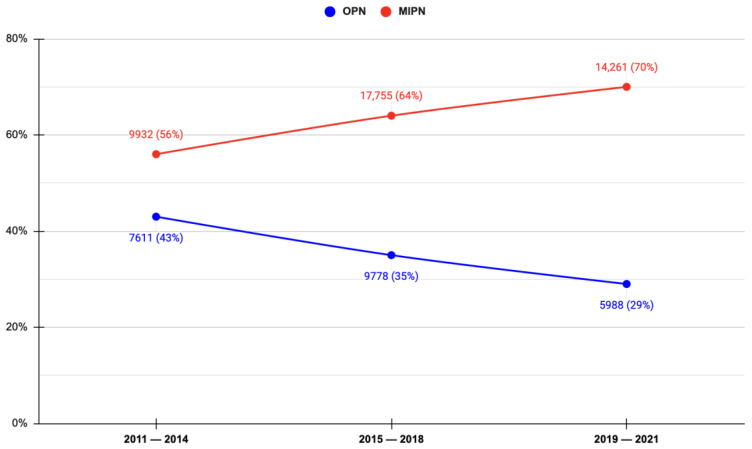
Temporal Trend of Open (OPN) vs. Minimally Invasive (MIPN) Partial Nephrectomy.

**Table 1 jcm-13-05454-t001:** Partial Nephrectomy Demographics.

	OPN	MIPN	*p* Value
	n (%)	n (%)	
Total	23,377	41,948	
Age, years			<0.001 *
Mean ± SD	60.6 ± 14.3	61.5 ± 11.8	
Sex, n (%)			=0.001 ^
Female	9209 (39%)	17,068 (40%)	
Male	14,168 (61%)	24,880 (60%)	
Region, n (%)			<0.001 ^
Midwest	6364 (28%)	12,251 (29%)	
Northeast	4941 (21%)	9255 (22%)	
South	9194 (40%)	14,889 (36%)	
West	2773 (11%)	5400 (13%)	
Unknown	105 (0.01%)	153 (0.01%)	
Insurance, n (%)			<0.001 ^
Private	15,925 (68%)	29,066 (69%)	
Medicaid	1296 (6%)	2173 (5%)	
Medicare	5738 (25%)	9769 (24%)	
Other Gov	264 (1%)	431 (1%)	
Cash	45 (0.01%)	77 (0.01%)	
Unknown	109 (0.01%)	432 (1%)	
CCI			<0.001 *
Mean ± SD	4.00 ± 3.29	4.48 ± 2.85	
≥3	18,508 (79%)	31,482 (75%)	<0.001 ^
Complications, n (%)			<0.001 ^
≤60 days	3621 (15%)	3980 (9%)	
Readmission, n (%) ≤30 days	1870 (8%)	2097 (5%)	<0.001 ^
SDOH, n (%)			<0.001 ^
≥1≤5 domains	3740 (16%)	4614 (11%)	
LoS			<0.001 *
Mean ± SD	4.78 ± 2.34	3.12 ± 1.82	
Year of Surgery			<0.001 ^
2011–2014	7611 (32%)	9932 (24%)	
2015–2018	9778 (42%)	17,755 (42%)	
2019–2021	5988 (26%)	14,261 (34%)	

Legend: * Student’s *t*-test; ^ chi-square test; Abbreviations: OPN (Open Partial Nephrectomy); MIPN (Minimally invasive Partial Nephrectomy; CCI (Charlson comorbidity index); SDOH (Social Determinants of health); LoS (Length of hospital stay).

**Table 2 jcm-13-05454-t002:** Breakdown of specific complications between OPN and MIPN.

Variable	OPN (n = 23,377)	MIPN (n = 41,948)	*p* Value
AKI	936 (4)	977 (2)	**<0.001**
Blood Transfusion	401 (2)	219 (0.5)	**<0.001**
DVT	267 (1)	230 (0.5)	**<0.001**
Leakage	119 (0.5)	109 (0.2)	**<0.001**
Ileus	150 (0.6)	231 (0.5)	0.143
Pneumonia	420 (2)	618 (1)	**0.001**
Pulmonary embolism	174 (0.7)	279 (0.6)	0.242
Sepsis	170 (0.7)	251 (0.5)	0.06
UTI	868 (4)	958 (2)	**<0.001**
Wound dehiscence	101 (0.4)	87 (0.2)	**<0.001**
Vascular Injury	15 (0.1)	21 (0.05)	0.461
Overall	3621 (15)	3980 (9)	**<0.001**

Abbreviations: OPN: open partial nephrectomy; MIPN: minimally invasive partial nephrectomy; Acute kidney injury (AKI); Deep Venous Thrombosis (DVT); Urinary Tract Infection (UTI).

**Table 3 jcm-13-05454-t003:** Multivariable Logistic Regression Analysis for Perioperative Complication Rates, N = 65,325.

	Adjusted Odds Ratio	95% Confidence Interval
Age	**1.01**	**1.01–1.02**
CCI	**1.10**	**1.09–1.11**
Type of approach		
MIPN	(Reference)	(Reference)
OPN	**1.62**	**1.54–1.70**
SDOH		
MIPN		
No SDOH	(Reference)	(Reference)
SDOH	**1.31**	**1.18–1.46**
OPN		
No SDOH	(Reference)	(Reference)
SDOH	**1.11**	**1.01–1.25**
Region		
Midwest	(Reference)	(Reference)
South	0.98	0.91–1.07
Northeast	0.95	0.87–1.05
West	0.94	0.84–1.05
Unknown	0.76	0.38–1.34
Plan		
Cash	(Reference)	(Reference)
Private	1.08	0.46–2.51
Medicaid	1.46	0.62–3.44
Medicare	1.13	0.49–2.64
Other Gov.	0.98	0.39–2.44
Unknown	1.26	0.51–3.11
Sex		
Female	(Reference)	(Reference)
Male	0.96	0.85–1.06
Year		
2011–2014	(Reference)	(Reference)
2015–2018	1.07	0.88–1.30
2019–2021	1.05	0.86–1.28

Abbreviations: CCI (Charlson comorbidity index); OPN (Open Partial Nephrectomy); MIPN (Minimally invasive Partial Nephrectomy; SDOH (Social Determinants of health).

## Data Availability

The data presented in this study are available on request from the corresponding author due to privacy policies.

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
