# Peer review of "Open versus Minimally Invasive Partial Nephrectomy: Trends and Outcomes from a Wide National Population-Based Database"

_jcm, 2024, doi:10.3390/jcm13185454_

Round 1
Reviewer 1 Report
Comments and Suggestions for Authors
Dear authors,
First of all, I would like to congratulate you on this theme. Although intensively discussed in the last decade, the parallel between OPN and MIPN continues to arouse the interest of urologists, oncologists and nephrologists. From this point of view, as well as from the perspective of a population study with a large number of patients, your article has an undeniable value.
Obviously, as in other studies, MPIN was associated with shorter length of hospital stay, less blood loss and complications, and also better preservation of renal function.
Back to the article:
The introduction is satisfactory, although I would have described the comparison techniques in more detail
The material and method section are concise, and the results section is correctly presented. Unfortunately, table 2 does not show significant differences in favor of MIPN, although the literature cites them
- I appreciated the fact that you presented the limitations related to the coding for laparoscopic and robot-assisted procedures. Indeed, it is difficult to differentiate only by code, retrospectively, which procedure it is about.
I'm wondering what do you want to say in conclusion section. You told me about MIPN, that gradually replaced OPN, which carries a higher risk of complication. But you proved more complications in table 2 for MIPN versus OPN (41,948 versus 23,377).
Good luck!
Reviewer 2 Report
Comments and Suggestions for Authors
The present manuscript seeks to demonstrate trends comparing open versus minimally invasive partial nephrectomy
While this topic is interesting, the data are very confirmatory and not surprising.
Specific criticisms:
Unfortunately, the outcome regarding patients with chronic kidney disease is neither addressed nor mentioned. This may play also a role in preferring the operating technique.
The outcome regarding kidney cancer should be detailed more clearly
It is not clear what kind of database they used. A US-data base? The description is very superficial. How are different ethinicities are represented therein, in the trends seen?
It should be more specifically discussed the differencies, if there any, between e.g Europe and the US regarding the trends, outcomes, insurance quality and so on.
Reviewer 3 Report
Comments and Suggestions for Authors
In this study, the authors investigated temporal trends and overall complication rates among open partial nephrectomy (OPN) and minimally invasive partial nephrectomy (MIPN), including impact of social 18 determinants of health (SDOH) on postoperative outcomes.
They concluded that MIPN is gradually replacing OPN, which carries a higher risk of complication, and SDOH are significant predictors of postoperative complications following PN, regardless of the approach.
This article is very interesting.
However, there is a major problem with this article.
The authors should include a table comparing the average values of surgical time, arterial ischemic time, bleeding volume, and postoperative estimated glomerular filtration rate (eGFR) between OPN and MIPN groups.
Reviewer 4 Report
Comments and Suggestions for Authors
Dear authors,
1. The manuscript mentions that the COVID-19 pandemic may have contributed to the reduction in surgical procedures between 2019 and 2021. However, a more detailed analysis of the temporal variations in surgical volumes before and after the pandemic, along with an examination of other influencing factors, could provide a deeper understanding of COVID-19's impact. Exploring these variables could offer more comprehensive insights into the pandemic's effects on surgical trends.
Lack of Long-Term Renal Function Data:
2. The absence of long-term renal function data and information on changes in GFR limits the conclusions regarding differences in renal function between MIPN and OPN. Future studies should aim to collect and analyze long-term data on GFR and postoperative recovery to better understand the impact of different surgical approaches on renal function over time.
Effectiveness of Robotic-Assisted Surgery:
3. The manuscript discusses the utility of robotic-assisted partial nephrectomy (RAPN) and cites the ROBOCOP II study. However, additional randomized controlled trials (RCTs) are needed to further clarify the benefits and limitations of RAPN. More research is required to establish a clearer understanding of RAPN's advantages and its place in surgical practice.
Impact of Social Determinants of Health (SDOH):
4. The discussion on the impact of social determinants of health (SDOH) on surgical choices and outcomes is insightful. To enhance the practical relevance of this research, it would be beneficial to include specific recommendations for addressing these disparities. Suggestions for policy interventions or community-based programs aimed at mitigating the effects of SDOH could strengthen the study's implications for improving healthcare equity.
Database Limitations and Selection Bias:
5. The manuscript acknowledges limitations and potential selection bias related to the Pearldiver Mariner database. To improve the reliability of the findings, it would be helpful to propose concrete strategies for addressing these issues. For example, comparing data with other databases or conducting a more detailed analysis of the impact of different insurance plans could help mitigate potential biases and provide a more accurate picture of the outcomes.
Reviewer 5 Report
Comments and Suggestions for Authors
Congratulations on a very interesting and well design study. The data is well presented and actually surprising: the time tranding on partial nephrectomy approaches is surprising!
I believe the design is easy to understand and data supports the results and conclusion well. There are no major issues on grammar or writing.
Originality can be argued, since similar papers have been published in the past. But overall, this has a very broad database and, for that reason, has a stronger merit.
Round 2
Reviewer 2 Report
Comments and Suggestions for Authors
The authors have just yet responded to the concerns raised.
Reviewer 3 Report
Comments and Suggestions for Authors
The authors responded to my suggestions.
This paper may be published as is.